# Graft Failure after Uterus Transplantation in 16 Recipients: A Review

**DOI:** 10.3390/jcm12052032

**Published:** 2023-03-03

**Authors:** Iori Kisu, Risa Matsuda, Tetsuro Shiraishi, Ryoma Hayashi, Yusuke Matoba, Masato Tamate, Kouji Banno

**Affiliations:** 1Department of Obstetrics and Gynecology, Keio University School of Medicine, Tokyo 160-8582, Japan; 2Department of Obstetrics and Gynecology, Sapporo Medical University, Sapporo 060-8543, Japan

**Keywords:** graft failure, hysterectomy, thrombosis, uterus transplantation, uterine factor infertility, uterine infection

## Abstract

Uterus transplantation (UTx) is now an alternative to surrogacy and adoption for women with uterine factor infertility to have children; however, there are still unresolved clinical and technical issues. One of these is that the graft failure rate after transplantation is somewhat higher than that of other life-saving organ transplants, which is a critical concern. Herein, we summarize the details of 16 graft failures after UTx with living or deceased donors using the published literature in order to learn from these negative outcomes. To date, the main causes of graft failure are vascular factors (arterial and/or venous thrombosis, atherosclerosis, and poor perfusion). Many recipients with thrombosis develop graft failure within one month of surgery. Therefore, it is necessary to devise a safe and stable surgical technique with higher success rates for further development in the UTx field.

## 1. Introduction

Uterus transplantation (UTx) is a new possible approach for women with uterine factor infertility (UFI) to have children. Following the first successful surgical engraftment of the uterus with a deceased donor by Ozkan et al. in Turkey in 2011 [1,2], the first successful live birth with a living donor was reported by Brännström et al. in Sweden in 2014 [3]. Moreover, the first successful live birth with a deceased donor was reported by Ejzenberg et al. in 2017 [4]. These remarkable achievements have attracted significant attention worldwide and this new procedure has become an alternative to surrogacy and adoption for women with UFI, especially those with Mayer–Rokitansky–Küster–Hauser (MRKH) syndrome, which occurs in 1 in 4500 women. To date, 84 UTx procedures have been performed worldwide with 49 newborns [5]. Although UTx is still in the experimental stage with many medical, technical, and ethical issues to be resolved [6,7,8], the efficacy of UTx is likely to be established as a treatment for women with UFI by the great achievements of the delivery of healthy babies after UTx. 

However, regarding the safety of this procedure, there are still open questions. In particular, living donor surgery is highly invasive because of the difficulty in procuring deep uterine veins running along the pelvic floor, which may result in a prolonged operation, significant bleeding, and surgical complications [8]. The rate of postoperative complications among living donors requiring surgical, endoscopic, or radiological interventions is high (18%), with the majority of the complications being injuries of the urinary tract [9]. Thus, an extremely precise surgical procedure involving dissections of deep uterine and internal iliac veins is required, preserving the vesical nerve branches of the hypogastric nerve and inferior hypogastric plexus in order to prevent postoperative dysuria [10]. 

In recipients, there are various physical and psychological burdens related to oocyte collection, transplant surgery, immunosuppressant use, etc., before and after UTx, including during pregnancy. Therefore, recipients must give full informed consent prior to participation in clinical trials because they may ultimately be unable to conceive through UTx despite these burdens. Moreover, anastomosis of vessels represents another challenging step because of the small vascular sizes, which leads to increased warm ischemia times and risks of graft explantation when vascular thrombosis occurs. In fact, the initial report from the International Society of Uterus Transplantation (ISUTx) stated that transplant hysterectomies were performed in 12 of 45 recipients (graft failure rate, 27%) [11]. This high graft failure rate is the most serious problem associated with UTx and UTx cannot become a universal medical technology unless a safe operation with a high success rate is established. Therefore, it is essential to further develop UTx to investigate the causes of graft failure and take countermeasures against this problem.

This study aimed to thoroughly review data on graft failures in recipients after UTx. Graft failure in UTx is defined as the need to unexpectedly explant the uterine graft before embryo transfer.

## 2. Materials and Methods

We searched the PubMed database on 5 December 2022, using the search terms “uterus transplantation” OR “uterine transplantation”. We included prospective cohort studies, reviews, case reports, and the ISUTx registry report that focused exclusively on human studies, used the English language, and published between March 2002 and December 2022. A manual search of bibliographies was independently conducted by two reviewers (I.K. and R.M.) and all data regarding graft failure after UTx were extracted.

## 3. Results

A total of 16 graft failures after UTx were reported by five centers in peer-reviewed journals as shown in Table 1: one in Saudi Arabia, five in Sweden, one in Cleveland, US, six in Dallas, US, and three in the Czech Republic. Several additional cases were presented at the ISUTx congress; however, their data have not been published or included in this review.

Vascular thrombosis was the most common cause of uterine graft failure. Graft failure occurred in most cases within one month after UTx. Hysterectomy was performed in all the cases (Table 1). Of the 16 recipients, 11 were living donors and the rest were deceased donors. Surgical approaches for the 11 living donors were robotic and open surgeries in two and nine cases, respectively (Table 2). 

Details of the graft failures after UTx experienced by each team are summarized.

### 3.1. Saudi Arabia

The first attempt at performing UTx was made by a team in Saudi Arabia in 2000 [12]. The recipient was a 26-year-old female who had lost her uterus six years ago due to postpartum hemorrhage and the living donor was a 46-year-old patient with multiloculated ovarian cysts. In case #1, the uterine arteries and veins were extended using reversed segments of the great saphenous vein to extend the length of the vascular pedicles. On postoperative day (POD) 99, the recipient experienced a sudden feeling of heaviness with a foul-smelling vaginal discharge. Unfortunately, the transplanted uterus prolapsed into the vagina with a dusk-colored cervix on speculum examination. Acute vascular thrombosis in the uterine arteries, veins, and supplying grafts occurred, resulting in uterine infarction. A hysterectomy was then performed. Uterine prolapse after UTx led to an emphasis on adequate fixation of the uterus at the pelvic floor in subsequent clinical applications. Furthermore, because it was a failure due to insufficient preparation, animal experiments were later extensively developed to prepare for clinical applications in humans [6,13].

### 3.2. Sweden

The first prospective cohort study was conducted by a Swedish team between 2012–2013 [3,14,15,16,17]. Nine recipients underwent UTx via open living-donor surgery. Two of the nine recipients (#2 and #3) developed graft failure after surgery. One recipient (#2) developed abdominal pain, fever, and vaginal discharge one month after the surgery and a cervical/uterine infection with a positive *Enterococcus faecalis* culture was confirmed. She was treated with antibiotics, but eventually developed an intrauterine abscess. Although surgical drainage was attempted, a hysterectomy was performed on POD 105. The explanted uterus mainly showed a viable myometrium, whereas the inner half was necrotic with focal neutrophil infiltration, but no signs of rejection [14,18]. The other recipient (#3) had acute bilateral thrombotic uterine artery occlusions, which was detected using a uterine artery Doppler signal with a blood-congested cervix on POD 3 and the graft was removed. Occluding thrombi were found in both major arteries, and focal necrosis and moderate ischemic myometrial damage were revealed by morphologic examination [14]. The recipient (#3) was heterozygous for protein C deficiency, which would increase the risk of venous thromboembolism; however, it is unclear whether the complication was related to this mutation. In this case, the team noted that other factors that may have predisposed the patient to thrombosis were that the transplant involved an older donor (62 years old) and that the post-anastomosis uterine arterial blood flow was the lowest among all the cases [14].

Between 2017 and 2019, the team conducted a second clinical trial using robot-assisted donor surgery in eight patients [19,20,21]. Two of the eight recipients (#4 and #5) developed graft failure with similar intercourse after surgery; transabdominal color Doppler ultrasound showed a decreased distribution of blood flow in the central portions, and initial biopsies around POD 4–5 revealed focal ischemia in the ectocervix [18,19]. In the first recipient (#4) with graft failure, bilateral thrombosis of the uterine arteries was observed during the UTx procedure after the initial anastomosis and reperfusion. Therefore, the anastomosis sites on the external iliac arteries were opened and thrombectomy was performed, resulting in a longer surgical time (6.6 h) and total graft ischemia time (3.2 h). After the surgery, restricted blood distribution was detected on color Doppler ultrasonography and the uterine size gradually decreased. Additionally, repeated cervical biopsies showed ischemia, followed by necrosis without any visible endometrium on hysteroscopy and transvaginal ultrasonography. Therefore, the atrophic uterus was explanted at eight months post-transplantation [18]. Regarding the second recipient (#5), the course and findings were similar to those of the first recipient (#4), with restricted blood distribution and ischemic changes; hysterectomy was performed one month after the operation based on the first experience. Both recipient #4 and #5 shared some similar features. Perioperative perfusion blood flow was lowest in both recipients, and infectious episodes emerged with *Escherichia coli* (#4) or *Enterococcus faecium* (#5) in a vaginal or cervical culture. Histopathologically, the endometrium and inner half of the myometrium were necrotic, whereas the outer half showed a partly viable myometrium [18]. Moderate atherosclerosis was observed in both the recipients; however, no signs of rejection were observed.

Notably, this team encountered a case of life-threatening post-transplant lymphoproliferative disorder after UTx with a deceased donor, thereby resulting in a hysterectomy. Recipient #6 had severe intestinal pain and bleeding episodes, and was diagnosed with Epstein–Barr virus-infected intestinal lymphoma [5,22]. Recurrent microperforations of the small bowel required several surgical procedures and embolization of the intestinal arteries. However, details of this case have not yet been published.

### 3.3. Cleveland (United States)

In the United States, a Cleveland team performed eight UTx procedures with a deceased donor [23,24,25,26,27]. The first case (#7) was conducted in 2016, which was the first attempt at a human UTx trial in the United States [27,28]. A stable initial postoperative course was achieved with bilateral patency of external iliac vessel anastomoses and graft viability on cervical biopsy. Unfortunately, an urgent hysterectomy was performed on POD 12 because of vascular Candida infection, which disrupted one of two arterial anastomoses and led to intra-abdominal bleeding. This type of infection has not been previously described in human or animal studies on UTx.

### 3.4. Dallas (United States)

A Dallas team started a UTx trial in 2016 and performed the most UTx procedures to date, with a total of 22 cases [5,23,29,30,31,32,33,34,35,36,37,38,39,40,41]. Of this total, 20 were UTx procedures performed by living donors and two by deceased donors [5,23]. Graft failure was reported in six (#8–13) of the 22 cases: five from living donors and one from a deceased donor. Although the team experienced consecutive graft failures in the first three cases (#8–10) of the trial, the negative consequences of the initial experience prompted a thorough analysis of each case [29].

The recipients (#8–10) lost their uterine grafts due to vascular complications related to inflow and outflow problems as the primary reason for their graft losses, all of which were performed with living donors [29,32]. Graft hysterectomy was performed on POD 14, 12, and six in recipient #8, #9, and #10, respectively. Notably, in all three recipients, vaginal Doppler ultrasounds showed blood flow in the uterine arteries and veins, despite an abnormal appearance of the cervix. In recipients #8 and #10, the transplanted cervices appeared congested, whereas in recipient #9, the cervix appeared ischemic. In the three cases, cervical biopsy showed necrotic changes and subsequent magnetic resonance imaging (MRI) confirmed graft necrosis, which resulted in graft hysterectomy. Pathological examination of recipients #8 and #10 demonstrated necrotic hemorrhagic damage suggestive of venous outflow problems. Recipient #9 showed ischemic necrosis with almost complete occlusion of the arterial lumen due to fibrosis of the arterial intima and arteriosclerosis [29]. Physical and systemic signs of necrosis were absent in all three cases. The team learned from the complications that MRI or direct surgical exploration may be necessary to avoid delayed recognition of graft necrosis due to vascular complications when perfusion within the body of the uterus cannot be confirmed on ultrasonography, when the cervix appears compromised on examination, or when a biopsy shows signs of necrosis. Based on this experience, the team modified the venous anastomosis technique to prevent thrombus formation. Thus, venotomy was performed more towards the medial wall of the iliac vein than on the superior wall of the iliac vein, which allowed for a more natural course for the outflow. Moreover, an oval orifice resection of a portion of the iliac veinous wall, instead of a simple slit, was performed to maximize outflow [29].

Of the three remaining graft failures in the Dallas trial (#11–13), only recipient #12 underwent UTx from a brain-dead donor. In recipients #11 and #12, the graft failed to properly reperfuse, despite patent vasculature [32]. In recipient #13, the transplant was technically successful, but the recipient experienced hemorrhagic shock immediately after surgery due to arterial bleeding. This led to irreversible ischemic damage to the graft, which was removed on POD 1 [32]. In the Dallas trial, no graft failures were observed in UTx using robot-assisted donor surgery.

### 3.5. Czech Republic

A Czech Republic team performed 10 UTx procedures between 2016 and 2018 with five living and five deceased donors in a two-arm study comparing the efficacy of UTx [42,43,44,45,46,47,48,49]. Graft failure occurred after the surgeries of three of the 10 recipients in this trial (#14–16) [43,49]. Early graft loss occurred in two recipients (#14 and #16) and mid-term graft loss occurred in one recipient (#15). In the first case of graft failure (#14), the cervix was congested and livid on visual appearance after surgery and the uterine blood flow ceased on Doppler ultrasound. A hematoma was removed and the right uterine artery was re-anastomosed on POD 1. However, direct removal of the graft was eventually carried out on POD 7 because of the necrotic appearance of the transplanted uterus during laparotomy. The team speculated that arterial thrombosis may have occurred according to explant histology, partly due to atherosclerosis.

The second case (#15) of graft failure required hysterectomy, despite a technically successful operation. Herpes simplex virus-2 (HSV-2) was detected in a uterocervical biopsy on POD 40; this was treated with antiviral therapy. However, no menstrual bleeding appeared without the endometrium on an ultrasound examination. Post-infection stenosis of the cervical canal was present and transcervical access to the uterine cavity was not possible. Despite hormonal therapy, endometrial growth was not observed. Therefore, a graft hysterectomy was performed on POD 213. Histopathological examination of the graft revealed fibrous obliteration of the cervical canal and uterine cavity, possibly due to a combination of chronic rejection changes (not found in a previous cervical biopsy), vascular thrombosis, graft atherosclerosis, and HSV infection.

The third case (#16) also underwent hysterectomy due to vascular thrombosis. Clinical symptoms similar to those of the first case (#14) (cervical congestion and blood flow disruption) were observed. Relaparotomy was performed on POD 5 and a congested but pink uterus with dilated veins was found. A hematoma was removed and thrombectomy of the left uterine vein was performed, followed by additional anastomosis of the right ovarian vein. Unfortunately, the cervix became livid again and Doppler vascular flow ceased. Hysterectomy was removed 15 days after the first UTx procedure. Histopathological examination revealed hemorrhagic necrosis of the graft. Additionally, immunological testing confirmed a positive T cell fluorescence-activated cell sorter crossmatch and the presence of de novo donor-specific antibodies against human leukocyte antigen class I, leading the team to speculate that thrombosis development involved immune rejection.

## 4. Discussion

Uterus transplantation is now an alternative to surrogacy and adoption for women with UFI desiring to have children; however, there are remaining clinical and technical issues yet to be resolved [7,8,10,50]. One of these is that the graft failure rate after transplantation is somewhat higher than that of other life-saving organ transplants, which is a critical concern. Therefore, identifying the causes of graft failure is important for further development of safe UTx surgeries. Furthermore, long-term psychological and medical follow-ups of donors and recipients are necessary because recipients may ultimately be unable to conceive through UTx due to graft failure. In this review, we summarized the details of graft failure after UTx with living or deceased donors using the published literature in order to learn from these negative outcomes.

A total of 16 graft failures after UTx have been reported by multiple facilities in the peer-reviewed literature. These facilities, with the exception of Saudi Arabia, have performed at least eight uterine transplants. In addition to the report in this review, one case each from China (Guangzhou) [51], Cleveland, and Brazil were presented at the ISUTx congress, however, detailed data have not been published. To the best of our knowledge, based on the published studies, presentations at the ISUTx congress, press releases, and personal communications, clinical application of UTx has been conducted for 96 patients, 71 from living donors and 25 from deceased donors. As for the graft failure rate, including the three unpublished cases mentioned above, the overall graft failure rate was 19.8% (19/96), 16.9% (12/71) of UTx from living donors and 28% (7/25) from deceased donors, indicating a higher graft failure rate for UTx from deceased donors. Of the 12 graft failures in living donor surgery, nine were open, two were robotic, and one was laparoscopic surgery.

The term “graft failure” in this article is defined as an unexpected explant of the uterine graft before embryo transfer and does not include cases of hysterectomy after delivery or hysterectomy after engraftment without resulting in pregnancy. Decision-making and the timing of removal in these situations are other concerns [2,52,53]. In particular, the exit strategy of hysterectomy is controversial when pregnancy is not achieved after uterine engraftment. In fact, the first trial by the Swedish team involved transplant hysterectomy approximately six years after UTx because of six miscarriages with 16 unsuccessful embryo transfers [15]. The purpose of UTx is to give birth and this reproductive failure may be treated as a graft failure in a narrow sense. In contrast, the Turkish team had five miscarriages, but finally achieved delivery nine years after UTx [2].

In the 16 cases, most graft failures were due to vascular causes (arterial and/or venous thrombosis, atherosclerosis, and poor perfusion). Many recipients with thrombosis developed graft failure within one month of surgery. Other major causes are infections, such as Candida and herpes infections. Graft failures related to rejection were rare and discovered after graft hysterectomy. Approximately one-third of UTx procedures are performed with deceased donors, but the graft failure rates for living and deceased donors appear to be comparable, as five out of 16 graft failures were from deceased donors. In living donor surgery, the uterine arteries and deep uterine veins are preserved, including segments of the internal iliac artery and vein. Dissection of the bilateral deep uterine veins, with their many branches and close proximity to the ureters, is complex and time-consuming. Therefore, the use of ovarian or utero-ovarian veins instead of uterine veins for the venous outflow has been attempted. Since UTx with living donors requires a smaller vascular anastomosis than renal or liver transplantation, the cause of intravascular thrombus may have a higher technical component. However, UTx from deceased donors does not appear to fall under this category because larger vessels can be procured. Intravascular thrombus was a more common cause of graft failure in UTx procedures in living donors than those in deceased donors (Table 1). Therefore, especially for UTx from living donors, measures must be taken to prevent thrombus formation, inadequate uterine perfusion, and venous congestion during vascular extraction and anastomosis.

Complications of intravascular thrombi can be divided into venous and arterial thrombi, and the countermeasures for each are likely to differ. To prevent complications of venous thrombosis, it is necessary to consider anastomotic techniques that do not cause blood flow congestion. Based on the experience with negative results in the first three cases associated with the Dallas team, they modified their venous vascular anastomosis technique and better results were obtained in subsequent cases [29,32,38]. In the arteries, the presence of atherosclerosis is associated with a high rate of arterial thrombosis and thrombus formation, and the resulting organ blood flow deficiency can lead to graft failure [54]. Therefore, patient selection will be important to ensure that those with severe atherosclerosis are not enrolled. However, despite the importance of preoperative imaging evaluation of high-quality uterine vasculature using computed tomography, digital subtraction angiography, and magnetic resonance angiography [55,56,57,58], it is often difficult to accurately assess the presence of atherosclerosis and vessel diameter using only preoperative imaging studies. Although a German team performed a thorough preoperative imaging evaluation of the donor’s uterus, they determined that adequate organ perfusion from the uterine artery could not be achieved during back table preparation, potentially creating a risk of failure. Therefore, they aborted transplant surgery in the prospective recipient [59,60]. In the Czech Republic team, the transplantation procedure was aborted because of diminutive uterine veins and fragile utero-ovarian veins at the back-table [49], but not with regard to the arteries. Thus, preoperative imaging evaluation, as well as intraoperative evaluation of vascular and perfusion status during back table preparation is important to avoid graft failure. Furthermore, whether donor lipid profiles and lipid-lowering therapy can affect the outcomes of UTx is unknown, but the effects on atherosclerosis progression and vascular quality may need to be considered. The impacts of preoperative donor age and endometrial thickness status on graft failure rate are also future issues.

A diagnosis of graft failure should be made promptly for subsequent correspondence of the recipient. Maintaining irreversibly unrecovered uterine function would impose unnecessary immunosuppressants, physical and psychological burdens, and risks on the recipient. While dysfunction in vital organs, such as the liver and kidneys, can often be detected by laboratory examinations, the uterus is not a vital organ and therefore, has no characteristic findings in laboratory studies. In cynomolgus experiments, it has been reported that blood cells, lactate dehydrogenase, and C-reactive protein are temporarily elevated in individuals with irreversible rejection [61], however, this is unknown in cases of graft failure due to vascular thrombosis. Symptoms of recipients with graft failure included lower abdominal pain and abnormal vaginal discharge, however, most were asymptomatic (Table 1). Therefore, the visual appearance of the cervix, uterine blood flow on Doppler ultrasonography, and pathological findings of the uterine biopsy are feasible means of diagnosing graft failure. Regarding the visual appearance of the cervix, cervical congestion and livid color are of note. On color Doppler ultrasonography, graft failure should be considered when there is no distribution of blood flow in the central portion of the uterus, even if the uterine vessel flow is good. Uterine atrophy and endometrial defects are also irreversible findings, in which case, a cervical biopsy should be performed to confirm the presence of ischemic changes [18,19,29].

In all 16 cases of graft failure, the transplanted uteri were excised. A major concern with UTx is whether a uterus with graft failure should be removed. Vital organs experience life-threatening conditions if their function is lost, however, the uterus is not immediately critical for survival, even when graft failure develops, because the uterus is not a life-saving organ. In renal transplantation, the rate of surgical transplant nephrectomy after graft failure (atrophy and/or possible necrosis) has been reported to be 20–80%, and it primarily depends on institutional policies [62]. There is no consensus on the timing and indications for allograft nephrectomy, with some studies suggesting that asymptomatically failed allografts should be removed because of the morbidity and mortality associated with transplant nephrectomy. Other studies suggest that failed allografts should be routinely removed because they can cause sepsis or chronic inflammation that can lead to complications [63,64]. Animal studies and clinical trials in humans have shown that a uterus with graft failure due to inadequate blood flow loses its endometrium and undergoes atrophy [19,61,65,66,67]. Therefore, it may be optimal not to perform an atrophied graft hysterectomy; however, a uterus with inadequate blood flow may be susceptible to infection because the uterine cavity is in contact with the outside of the body through the vagina, whereas the kidneys and liver are not in direct contact with the body. In fact, in the first Swedish trial of UTx, the recipient (#2) had repeated intrauterine infections after UTx and the uterus was resected [14].

It is expected that more than 100 cases of UTx will be performed worldwide in the near future, and new technology has the potential to break out of the experimental stage and become a standard treatment for women with UFI. However, the engraftment rate after transplantation is still lower than that of other organs. Therefore, it is essential to clarify and verify the negative data to overcome this problem.

In conclusion, to date, the main causes of graft failure have been vascular factors (arterial and/or venous thrombosis, atherosclerosis, and poor perfusion). Many recipients with thrombosis develop graft failure within one month of surgery. Therefore, it is necessary to devise safe and stable surgical techniques with a higher success rate for further development in the UTx field.

## Figures and Tables

**Table 1 jcm-12-02032-t001:** Details of graft failure cases after uterus transplantation.

Case	Country	Graft Survival Periods	Donor Type	Cause of Graft Failure	Symptoms or Trigger for Graft Failure	Procedure for Uterine Graft
#1	Saudi Arabia	99 days	LD	Vascular thrombosis	A sudden feeling of heaviness, vaginal discharge, uterine prolapse	Hysterectomy
#2	Sweden	105 days	LD	Intrauterine infection	Abdominal pain, fever, vaginal discharge	Hysterectomy
#3		3 days	LD	Bilateral uterine arterial thrombosis	Cessation of the uterine artery Doppler signal	Hysterectomy
#4		8 months	LD	Ischemia, necrosis (atherosclerosis)	Restricted blood distribution by color Doppler, ischemic changes by biopsy	Hysterectomy
#5		1 month	LD	Ischemia, necrosis (atherosclerosis)	Restricted blood distribution by color Doppler, ischemic changes by biopsy	Hysterectomy
#6		N/R	DD	PTLD	Severe intestinal pain and bleeding	Hysterectomy
#7	Cleveland, United States	12 days	DD	Candida infection	Intraabdominal bleeding	Hysterectomy
#8	Dallas, United States	14 days	LD	Insufficient venous outflow	Congested cervix, necrotic changes by biopsy	Hysterectomy
#9		12 days	LD	Ischemic necrosis intimal fibrosis and atherosclerotic of arteries	Ischemic cervix, necrotic changes by biopsy	Hysterectomy
#10		6 days	LD	Insufficient venous outflow	Congested cervix, necrotic changes by biopsy	Hysterectomy
#11		<30 days	DD	Poor perfusion	N/R	Hysterectomy
#12		<30 days	LD	Uterine arterial thrombosis	N/R	Hysterectomy
#13		1 day	LD	Uterine vein thrombosis	Hemorrhagic shock	Hysterectomy
#14	Czech Republic	7 days	DD	Uterine arterial thrombosis (atherosclerosis)	Congested cervix, ceased Doppler vascular flow	Hysterectomy
#15		213 days	DD	HSV infection, chronic rejection	HSV infection on cervical biopsy, No growth of endometrium	Hysterectomy
#16		15 days	LD	Venous thrombosis (positive T-FACS cross match, de novo DSA)	Congested cervix, ceased Doppler vascular flow	Hysterectomy

DD, deceased donor; DSA, donor-specific antibodies; HSV, herpes simplex infection; LD, living donor; PTLD, posttransplant lymphoproliferative disorder; T-FACS, T cell fluorescence-activated cell sorter; N/R, not reported.

**Table 2 jcm-12-02032-t002:** Characteristics of recipients and donors for uterus transplantation.

Case	Country	Donor Type	Donor Age	Menopausal Status on Donor	Surgical Approach for LD	Recipient	Recipient Age
#1	Saudi Arabia	LD	46	Post	Open	PPH	26
#2	Sweden	LD	58	Post	Open	MRKH	38
#3		LD	62	Post	Open	MRKH	35
#4		LD	55	Post	Robotic	MRKH	33
#5		LD	46	Pre	Robotic	MRKH	23
#6		DD	N/R	N/R	N/A	MRKH	N/R
#7	Cleveland, United States	DD	N/R	N/R	N/A	MRKH	N/R
#8	Dallas, United States	LD	42	Pre	Open	MRKH	32
#9		LD	55	Post	Open	MRKH	33
#10		LD	45	Pre	Open	MRKH	34
#11		DD	44	N/R	Open	MRKH	29
#12		LD	48	Post	Open	Hysterectomy due to myoma	29
#13		LD	32	Pre	Open	MRKH	21
#14	Czech Republic	DD	57	Post	N/A	MRKH	29
#15		DD	56	Post	N/A	MRKH	32
#16		LD	49	Pre	Open	MRKH	25

DD, deceased donor; DSA, donor-specific antibodies; LD, living donor; MRKH, Mayer–Rokitansky–Küster–Hauser; N/A, not applicable; N/R, not reported; PPH, postpartum hemorrhage.

## Data Availability

Data available in a publicly accessible repository.

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
