# Peer review of "Graft Failure after Uterus Transplantation in 16 Recipients: A Review"

_jcm, 2023, doi:10.3390/jcm12052032_

Round 1

Reviewer 1 Report

The review “Graft Failure after Uterus Transplantation in 16 Recipients” summarize the details of 16 graft failures after uterus transplantation with living or deceased donors using published literatures in order to learn from these negative outcomes. To date, the main causes of graft failure are vascular factors (arterial and/or venous thrombosis, atherosclerosis, and poor perfusion).

Keywords: add “uterus infection”.

Introduction: adequate systematical reports of  the previous literature.

Results: the 16 cases of graft failures after uterus transplantation are described case by case. In particular the authors describe the donor type, cause of graft failure and symptoms for graft failure.

Discussion: needs to be expanded.  In particular, please describe:

1. the most appropriate surgical technique to date, adding explanatory figures

2. immunosuppressive therapy for graft failure (type and dosage of drugs, etc.)

Further,  add to the discussion the need for long-term psychological and medical follow-up of donors, recipients and children.

Author Response

Comment:

Keywords: add “uterus infection”.

Response:

As pointed out by the reviewer, we have added “uterine infection” to Keywords.

Comment:

Discussion: needs to be expanded.  In particular, please describe:

  1. the most appropriate surgical technique to date, adding explanatory figures

Response:

We thank the reviewer for this crucial comment. As pointed out by the reviewer, we have added the following descriptions in the section of Discussion.

“In living donor surgery, the uterine arteries and deep uterine veins are preserved including segments of the internal iliac artery and vein. Dissection of the bilateral deep uterine veins, with their many branches and close proximity to the ureters, is complex and time-consuming. Therefore, the use of ovarian or utero-ovarian veins instead of uterine veins for the venous outflow has been attempted.”

As for the insertion of figures, it is not the main purpose of this Review article, so we would like to omit it. We would be pleasured if the reviewer could understand.

Comment:

  1. immunosuppressive therapy for graft failure (type and dosage of drugs, etc.)

Response:

Since there are few cases of rejection among graft failure cases and, conversely, many cases of rejection among successful cases, it is unlikely that the method of administration of immunosuppressive agents has an effect on graft failure cases, and it is thought that the description of immunosuppressive agents may cause confusion in understanding. We would like to omit the description of immunosuppressive agents. Thank you for your understanding.

Comment:

Further, add to the discussion the need for long-term psychological and medical follow-up of donors, recipients and children.

Response:

We thank the reviewer for this crucial comment. As pointed out by the reviewer, we have added the following descriptions in the section of Discussion.

“Furthermore, long-term psychological and medical follow-up of donors and recipients are necessary because recipients may ultimately be unable to conceive through UTx due to graft failure.”

Reviewer 2 Report

Dear Authors, 

The idea of making this review is interesting and promising due to the still limited experience in this field.

Some aspects will have to be clarified.

-        Line 118 - heterozygous PC deficiency alone does not determine that a subject has thrombophilia. DOI: 10.1111/jth.14667

-        Was the preoperative preparation different at the level of the teams?

-        How can the donor's lipid profile and lipid-lowering therapy influence the atherosclerosis process and the quality of its vessels? HDL antioxidant function is significantly higher in patients with graft failure.

-        What were the indications for transplantation; it must be mentioned for which pathologies they were performed. 

-        Line 273, for better understanding mention what changes they made to lower the graft failure rate

-        The role of patient counseling in the process of performing the uterine transplant

-        To be discussed if there were differences regarding the immunosuppression, anticoagulation schemes between the teams

-        The failure rate is not only given by the failure of the uterine transplant from a surgical point of view, it is also given by the inability of the transplanted uterus to maintain a pregnancy. How can the thickness of the endometrium in the case of donors and its age influence the failure rate?

-        Line 321 not only atrophy but also possible necrosis

-        Since the indication was mostly for MRKH, it is important to mention type I or II of this condition.

-        How the duration of uterine transplant surgery can influence the failure rate?

Kind regards

Author Response

Comment:

Line 118 - heterozygous PC deficiency alone does not determine that a subject has thrombophilia. DOI: 10.1111/jth.14667

Response:

We thank the reviewer for this crucial comment and the reference. We have deleted this sentence to avoid misunderstandings.

Comment:

Was the preoperative preparation different at the level of the teams?

Response:

From the information in the published papers, we do not believe it is possible to compare or determine whether there were differences among the teams, especially in preoperative preparation.

Comment:

How can the donor's lipid profile and lipid-lowering therapy influence the atherosclerosis process and the quality of its vessels? HDL antioxidant function is significantly higher in patients with graft failure.

Response:

We thank the reviewer for this crucial comment. There are no papers discussing whether the donor's lipid profile and lipid-lowering therapy can influence the atherosclerosis process and the quality of its vessels, and we believe this is an important issue for the future. We have added the following sentence to the Discussion.

“Furthermore, whether donor lipid profiles and lipid-lowering therapy can affect the outcomes of UTx is unknown, but the effects on atherosclerosis progression and vascular quality may need to be considered.”

Comment:

What were the indications for transplantation; it must be mentioned for which pathologies they were performed.

Response:

We described the pathologies of the recipients in Table 2, which were almost MRKH syndrome.

Comment:

Line 273, for better understanding mention what changes they made to lower the graft failure rate

Response:

We described the measure by Dallas team in the next paragraph as below.

“Based on the experience with negative results in the first three cases by the Dallas team, they modified their venous vascular anastomosis technique, and better results were obtained in subsequent cases [29,32,38].”

The specific details were described in the text as follows.

“Based on this experience, the team modified the venous anastomosis technique to prevent thrombus formation. Thus, venotomy was performed more towards the medial wall of the iliac vein than on the superior wall of the iliac vein, which allowed a more natural course for the outflow. Moreover, an oval orifice resection of a portion of the iliac veinous wall, instead of a simple slit, was performed to maximize outflow [29].”

Comment:

The role of patient counseling in the process of performing the uterine transplant

Response:

As pointed out by the reviewer, we have added the following descriptions in the section of Discussion.

“Furthermore, long-term psychological and medical follow-up of donors and recipients are necessary because recipients may ultimately be unable to conceive through UTx due to graft failure.”

Comment:

To be discussed if there were differences regarding the immunosuppression, anticoagulation schemes between the teams

Response:

Since there are few cases of rejection among graft failure cases and, conversely, many cases of rejection among successful cases, it is unlikely that the method of administration of immunosuppressive agents has an effect on graft failure cases, and it is thought that the description of immunosuppressive agents may cause confusion in understanding. We would like to omit the description of immunosuppressive agents. Thank you for your understanding.

As for the mention of anticoagulant therapy, we also would like to refrain from mentioning it to avoid confusion, since it cannot be concluded that those coagulant therapies were inappropriate and directly led to the thrombus.

Comment:

The failure rate is not only given by the failure of the uterine transplant from a surgical point of view, it is also given by the inability of the transplanted uterus to maintain a pregnancy. How can the thickness of the endometrium in the case of donors and its age influence the failure rate?

Response:

We agree with the opinion by the reviewer. Therefore, we described the intention in the Discussion as follows; “In fact, the first trial by the Swedish team involved transplant hysterectomy approximately six years after UTx because of six miscarriages with 16 unsuccessful embryo transfers [15]. The purpose of UTx is to give birth, and this reproductive failure may be treated as a graft failure in a narrow sense. In contrast, the Turkish team had five miscarriages, but finally achieved delivery nine years after the UTx [2].”

It is not possible to conclude in this paper how endometrial thickness or age in the case of donors affects the failure rate. Therefore, the following text has been added to Discussion.

“The impact of donor age and endometrial thickness status on the failure rate is also a future issue.”

Comment:

Line 321 not only atrophy but also possible necrosis

Response:

As the reviewer pointed out, we have added the words, atrophy and/or possible necrosis as below.

“In renal transplantation, the rate of surgical transplant nephrectomy after graft failure (atrophy and/or possible necrosis) has been reported to be 20–80%”

Comment:

Since the indication was mostly for MRKH, it is important to mention type I or II of this condition.

Response:

We decided not to include this information because most reports do not include this information. We hope the reviewer understand.

Comment:

How the duration of uterine transplant surgery can influence the failure rate?

Response:

We do not think it is possible to conclude that the surgery time factor alone affects the failure rate, so we have omitted such a statement.

Reviewer 3 Report

“Graft Failure after Uterus Transplantation in 16 Recipients: A Review”

The manuscript presented for review consists of 12 pages with 68 references. 2 tables are included. Abbreviations used in the tables have been placed and explained in the legend. The authors focused on the issue concerning uterus transplantation and its consequences. The English language is understandable and appropriate. The work fits the journal scope (Section Obstetrics & Gynecology) - The Era of New Assisted Reproductive Technology (ART). The study is interesting for the reader. The study is correctly designed. The methods are described in sufficient details. The references are recent and adequate to the subject. 

I would suggest to describe Mayer-Rokitansky-Küster-Hauser (MRKH) syndrome in the introduction.

Author Response

Comment:

The manuscript presented for review consists of 12 pages with 68 references. 2 tables are included. Abbreviations used in the tables have been placed and explained in the legend. The authors focused on the issue concerning uterus transplantation and its consequences. The English language is understandable and appropriate. The work fits the journal scope (Section Obstetrics & Gynecology) - The Era of New Assisted Reproductive Technology (ART). The study is interesting for the reader. The study is correctly designed. The methods are described in sufficient details. The references are recent and adequate to the subject.

I would suggest to describe Mayer-Rokitansky-Küster-Hauser (MRKH) syndrome in the introduction.

Response:

We thank the reviewer for such a positive comment. We have added some description on MRKH syndrome in the introduction as the reviewer suggests.

Reviewer 4 Report

Comments and Suggestions for Authors

The manuscript by Kisu et al. represents an interesting body of research evidence in the clinical trials of Uterus Transplantation. The manuscript represented Uterus Transplantation failures in different trials from 2002 to 2022. However, some revisions must be done in this manuscript. 

L 62. In the Method section, the authors must declare how many studies there were in the search method and how many of them were chosen for the current study?

L 81. Table 1 must be edited. There are four horizontal lines in the body of the table. It must be deleted.

L 85. Table 2 also have the same issue as table 1. Moreover, the “PPH” abbreviation is not defined in the Table footnote. 

However, the tables 1 and 2 represent the data of transplantations very well, but as the number of studies is sufficient, it is more suitable to perform an analysis between the reasons of tissue rejection and show them on figures for better understanding.

L 103. The authors represented interesting data and details of patients with graft failure, however, it is suitable to talk about the success and eventual fertility status of the other surgeries, either. For example, how many of these successful surgeries led to a child born?

This data is important for total comparison of surgery methods represented in the current study.

L 152. Were the other seven transplantations successful? Please add the successful results of all studies presented in the current study.

I highly recommend that the authors visualize the important results and comparisons such as rate of graft failure of deceased and non-deceased donors, number of transplant failure in each surgery method (or surgery groups) etc. in figures for better understanding.

The authors stated the future prospective of the current study, however a clear conclusion is needed. A separate conclusion must be added into the manuscript.

Author Response

Comment:

L 62. In the Method section, the authors must declare how many studies there were in the search method and how many of them were chosen for the current study?

Response:

Since many cases in the same group are duplicated in multiple papers, reviews and ISUTx reports, we do not think it is significant to include the search strategy the reviewer mentioned, so we have included it in the form of “manual research”. We hope the reviewer will understand.

Comment:

L 81. Table 1 must be edited. There are four horizontal lines in the body of the table. It must be deleted.

Response:

We have deleted the four horizontal lines.

Comment:

L 85. Table 2 also have the same issue as table 1. Moreover, the “PPH” abbreviation is not defined in the Table footnote.

Response:

We have deleted the horizontal lines and added the footnote for PPH.

Comment:

However, the tables 1 and 2 represent the data of transplantations very well, but as the number of studies is sufficient, it is more suitable to perform an analysis between the reasons of tissue rejection and show them on figures for better understanding.

Response:

Since most graft failures are thrombosis and only one case was clearly diagnosed as rejection, we do not consider it appropriate to indicate the reason for rejection by number.

Comment:

L 103. The authors represented interesting data and details of patients with graft failure, however, it is suitable to talk about the success and eventual fertility status of the other surgeries, either. For example, how many of these successful surgeries led to a child born?  This data is important for total comparison of surgery methods represented in the current study.

Response:

As the reviewer states, the success rate of UTx surgery and subsequent pregnancy and delivery outcomes are important, but since the purpose of this review is to identify the causes of graft failure, we refrained from describing the success rate and other comparisons because they are not the main purpose of this paper. Instead, we have added a description of the graft failure rate as a response to the following comment. We appreciate your understanding.

Comment:

L 152. Were the other seven transplantations successful? Please add the successful results of all studies presented in the current study.

Response:

The true success of UTx is to obtain a child, but we do not think it is meaningful to show the successful results of only the team shown in this paper. Moreover, this definition would classify as unsuccessful even cases of pregnancy in which delivery is expected. For the same reason as the response to the one previous comment, we would like to refrain from describing them at this time. Instead, we have added a description of the graft failure rate as a response to the following comment. We would appreciate your understanding.

Comment:

I highly recommend that the authors visualize the important results and comparisons such as rate of graft failure of deceased and non-deceased donors, number of transplant failure in each surgery method (or surgery groups) etc. in figures for better understanding.

Response:

We appreciate the reviewer’s recommendation. we have added the following descriptions in the section of Discussion.

“To the best of our knowledge, based on the published studies, presentations at the ISUTx congress, press releases and personal communications, clinical application of UTx has been conducted for 96 patients, 71 from living donors and 25 from deceased donors. As for the graft failure rate, including the three unpublished cases mentioned above, the overall graft failure rate was 19.8% (19/96), 16.9% (12/71) of UTx from liv-ing donors and 28% (7/25) from deceased donors, indicating a higher graft failure rate for UTx from deceased donors.  Of the 12 graft  failures in living donor surgery, 9 were open, 2 were robotic, and 1 was laparoscopic surgery.”

Comment:

The authors stated the future prospective of the current study, however a clear conclusion is needed. A separate conclusion must be added into the manuscript.

Response:

We have added the conclusion of this manuscript in the end of the text.

Round 2

Reviewer 2 Report

I congratulate the authors for the work done because it is a future topic with implications not only in ensuring fertility in the case of patients with MRKH syndrome but also in other gynecological pathologies.

Kind regards